# Lupus Nephritis Subtype Classification with only Slide Level labels

**Amit Sharma**[*1]                                         AMIT.S@RESEARCH.IIIT.AC.IN

[1] *Center for Visual Information Technology, International Institute of Information Technology, Hyderabad, India*

**Ekansh Chauhan**[*1]                          EKANSH.CHAUHAN@RESEARCH.IIIT.AC.IN

**Megha S Uppin**[2]                                    MEGHA_HARKE@YAHOO.CO.IN

[2] *Department of Pathology, Nizam's Institute Of Medical Sciences, Hyderabad, India*

**Liza Rajasekhar**[3]                                 LIZARAJASEKHAR@GMAIL.COM

[3] *Department of Clinical Immunology and Rheumatology, Nizam's Institute Of Medical Sciences, Hyderabad, India*

**C V Jawahar**[1]                                            JAWAHAR@IIIT.AC.IN

**P K Vinod**[4]                                             VINOD.PK@IIIT.AC.IN

[4] *Center for Computational Natural Sciences and Bioinformatics, International Institute of Information Technology, Hyderabad, India*

**Editors:** Accepted for publication at MIDL 2024

## Abstract

Lupus Nephritis classification has historically relied on labor-intensive and meticulous glomerular-level labeling of renal structures in whole slide images (WSIs). However, this approach presents a formidable challenge due to its tedious and resource-intensive nature, limiting its scalability and practicality in clinical settings. In response to this challenge, our work introduces a novel methodology that utilizes only slide-level labels, eliminating the need for granular glomerular-level labeling. A comprehensive multi-stained lupus nephritis digital histopathology WSI dataset was created from the Indian population, which is the largest of its kind. *LupusNet*, a deep learning MIL-based model, was developed to classify LN subtypes. The results underscore its effectiveness, achieving an AUC score of 91.0%, an F1 score of 77.3%, and an accuracy of 81.1% on our dataset in distinguishing membranous and diffused classes of LN.

**Keywords:** Lupus Nephritis, Weakly Supervised Learning, Whole Slide Image, Binary Classification

## 1. Introduction

Lupus Nephritis (LN) is one of the most severe manifestations of systemic lupus erythematosus (SLE), an autoimmune disease, due to its potential for severe renal damage and the intricate diagnostic and classification process. The complex nature of this disease is worsened by the substantial-high inter and intra-observer variability in histopathological renal biopsies (Dasari et al., 2019). As some classes of LN exhibit varying levels of aggressiveness, a precise classification of these classes becomes crucial in assessing fatality risks, predicting long-term prognosis, and determining a practical therapeutic approach.

---

[*] Contributed equally

Deep learning has recently emerged as a powerful tool in medical AI and healthcare, revolutionizing various aspects of medicine, from diagnosis and treatment to drug discovery and patient monitoring (Rajkomar et al., 2018). Digital pathology has significantly advanced due to its capacity to extract intricate patterns and features from complex medical data (Wu and Moeckel, 2023; Ahmed et al., 2022). Improvements in image analysis have led to significant advancements in various aspects of renal pathology, including automated detection and classification of glomerular lesions (Sheehan and Korstanje, 2018; Ginley et al., 2019), and identification of interstitial fibrosis (Zheng et al., 2021a). Advanced imaging techniques and molecular analyses may assist, but standardization and consensus in interpretation remain ongoing challenges.

Traditional LN classification follows a two-step process: first, identifying glomeruli types, then classifying LN based on these types, heavily dependent on detailed glomeruli annotations (Sheehan and Korstanje, 2018; Zheng et al., 2021b). Yet, annotating glomeruli on large-scale WSIs is impractical in clinical settings due to their massive size and memory limitations, leading to patching and streaming solutions (Campanella et al., 2019; Pinckaers et al., 2020). Previous studies mainly differentiated LN from non-LN, not addressing subtype classification (Wang et al., 2023), which is complicated by similar glomerular types across subtypes and the unequal contribution of glomeruli to classification. (Cicalese et al., 2021) proposed an end-to-end LN subtype classification method, but it required manual segmentation on mice biopsies, not directly applicable to human samples due to differences in physiology and pathology.

In contrast, our work simplifies this process by creating an end-to-end pipeline that does not necessitate reliance on glomeruli class labels at any intermediate stage. Multiple Instance Learning (MIL) has been extensively explored for other areas of digital histopathology (Campanella et al., 2019), but not much has been reported or explored in renal pathology.

While digital pathology has made strides, the LN classification research faces challenges such as access to the datasets and lack of consensus among medical professionals regarding its classification. In light of these considerations, the principal contributions of our work are as follows:

- We focus on creating a valuable dataset of LN to drive research (computational and medical) in kidney diseases. This dataset, featuring multi-stained whole slide images, stands as one of the largest collections for lupus nephritis, a part of the consortium India Pathology Dataset (IPD) [1].

- We also introduce a novel architecture, LupusNet, an explainable MIL-based model that significantly improves LN subtype classification by integrating Gated and Multi-Head Attention, underscoring the critical requirement to learn the morphological differences between LN classes 4 & 5.

- To the best of our knowledge, we present the first end-to-end pipeline for LN subtype classification by relying only on slide-level labels, eliminating two-step methods that relied on glomeruli labels, easing clinical workload and facilitating practical integration.

---

1. https://hai.iiit.ac.in/ipd/

## 2. Materials and Method

### 2.1. Data Acquisition & Description

In this study, biopsy specimens of 166 patients (retrospective and prospective cases) in different subclasses (ranging from 1 to 6) of LN from the Nizam Institute of Medical Sciences (NIMS) in Hyderabad, India, were digitalized. A total of 540 WSIs were digitalized using the Morphle Optimus 6X Scanner, with each WSI captured at a maximum magnification of 40x and stored in the widely used TIFF format. Slide-level labels depicting subtype classes for each of the cases were also recorded.

Within this repository of 540 WSIs, there are four distinct categories of stained images, specifically Hematoxylin and Eosin (H&E), Periodic Acid-Schiff (PAS), methenamine silver Periodic Acid-Schiff (mt-PAS), and silver methenamine Periodic Acid-Schiff (sm-PAS). In this dataset, LN classes 4 (diffused proliferated) and 5 (membranous) exhibited the highest representation, with 62 and 53 cases, respectively. Class 4 LN displays a varied glomerular appearance characterized by widespread inflammation, cellular proliferation, and diverse lesions, whereas class 5 LN demonstrates a uniform appearance due to immune complex deposition, resulting in a membranous pattern (Weening et al., 2004). Figure 1 shows glomerulus samples from our dataset for each class. Consequently, our study focused primarily on observations and results for these two prominent LN class classifications using PAS-stained slides highlight carbohydrates, glycogen, and glycoproteins, aiding the identification of renal structures.

This India region-specific dataset is created to support global collaboration in lupus nephritis research. It helps add diversity to the other existing cohort, offering insights into potential regional and ethnic variations in the disease.

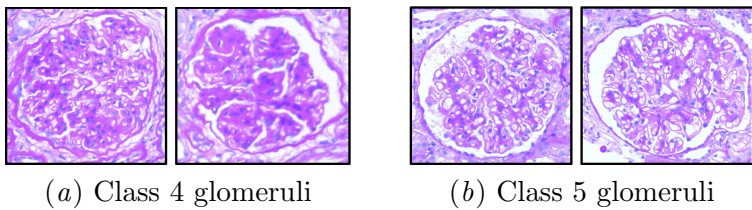

(*a*) Class 4 glomeruli      (*b*) Class 5 glomeruli

Figure 1: Comparison of visual features between subtype samples. (a) involves proliferative changes in the glomeruli, whereas (b) shows thickening of the glomerular basement membrane

### 2.2. Methodology

We aim to learn a function that can predict the presence or absence of a condition within a WSI based on its constituent patches. Mathematically, this problem can be defined as follows: We are provided with a dataset containing pairs of bag-labels $\{(X_i, Y_i)\}_{i=1}^{D}$. Each $X_i$ represents a collection of instances (patches) within a bag, and $Y_i$ is the label assigned to that bag. Each bag $X_i$ contains a variable number of instances $\{x_1, x_2, \ldots, x_N\} \in X_i$. These instances have labels $\{y_1, y_2, \ldots, y_N\}$ with $y_n \in \{0, 1\}$. However, the labels for individual

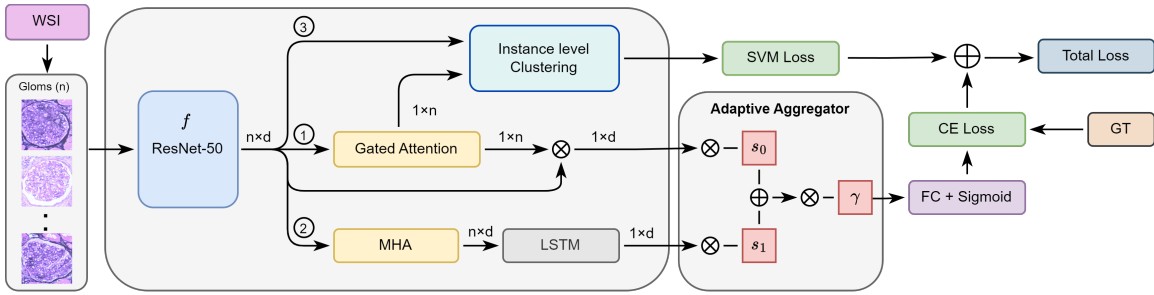

Figure 2: **LupusNet**: Proposed architecture for our lupus nephritis classifier. Gated attention identifies each glomerulus's importance, while multi-head attention (MHA) discerns their contextual relationships.

instances are unknown during the training phase. If any instance in a bag belongs to the positive class, then the bag is considered positive. Conversely, if all the instances in a bag belong to the negative class, the bag is considered negative.

$$Y_i = \begin{cases} 1, & \text{if } \exists x_n \in X_i \text{ such that } y_n = 1 \\ 0, & \text{otherwise} \end{cases}$$

Our methodology extends this formulation to multiple positive classes for subtype LN classification. Unlike lung, brain, and breast datasets, renal pathology primarily focuses on a limited region of interest, particularly the glomerular area, allowing us to use recurrent networks. Glomeruli play a pivotal role in various renal diseases, including LN. Instead of providing MIL with all WSI patches, we exclusively use glomerular patches, enhancing precision by avoiding potential noise. Recognizing the laborious labeling at the glomerular area, we aimed to eliminate the need for intermediate glomerular-level labels; thus, opting for weakly supervised approaches is an appropriate option.

Our novel end-to-end MIL architecture for LN classification, LupusNet, works on raw glomerular patches extracted using a fine-tuned YOLOv4 model (Hemmatirad et al., 2023), with two key components: (a) Feature Extractor ($f$) and (b) Feature Aggregator ($g$), jointly trained. $f$ transforms inputs into an information-rich feature space using a ResNet-50 network pre-trained on histopathology images (Kang et al., 2023). We built on CLAM principles (Lu et al., 2021), which utilizes gated attention pooling and instance-level clustering to distinguish positive from negative samples. Gated attention, however, cannot fully exploit the uniformity of class 5 lupus nephritis glomeruli, hindering its ability to achieve optimal efficacy in capturing its consistent patterns. We hypothesize that adding contextual information among all glomeruli patches will improve the performance. To address this, we integrate self-attention and Bi-LSTM into the MIL framework, enhancing contextual understanding among instances (patches) in a WSI.

Suppose, in a WSI bag $X$, we have $N$ glomerular patches, and the Feature Extractor $f$ transforms each image $x_n \in \mathbb{R}^{224 \times 224 \times 3}$ into a $h$ vector of dimension $d \in \mathbb{R}^{1 \times d}$. For $N$ such images, we obtain a matrix $H \in \mathbb{R}^{N \times d}$ (eq: 1). Our feature aggregator can further

be divided into three branches: (1) Gated Attention Pooling, (2) Self-Attention + LSTM and (3) Instance-level Clustering. In Branch 1, the gated attention block assigns attention scores $A^g = \{a_1^g, a_2^g, \ldots, a_N^g\} \in \mathbb{R}^{1 \times N}$ to every instance (eq: 2), followed by instance-level clustering using $A^g$ as pseudo labels for confident instances (Branch 3).

$$H = f(X; \Theta) \quad \text{where } H = \{h_1, h_2, \ldots, h_N\} \tag{1}$$

$$a_k^g = \frac{W_c^T(\tanh(W_a h_k^T) \odot \sigma(W_b h_k^T))}{\sum_{j=1}^{N} W_c^T(\tanh(W_a h_j^T) \odot \sigma(W_b h_j^T))} \tag{2}$$

$$C^g = \sum_{k=1}^{N} a_k^g h_k \tag{3}$$

where $W_a, W_b$ and $W_c$ are trainable parameters, $a_k^g$ can be supposed as positive probability of instances. $\sigma$ represents sigmoid function and $\odot$ represents element-wise multiplication. $C^g$ is the output context vector of Branch 1 (eq: 3).

In Branch 2, initially, $H$ goes to MHA, yielding contextualized output among instances ($A^s$). Self-attention (eq: 4) enables context consideration between every instance pair, and the multi-head mechanism focuses on modeling various such contextual relationships and dependencies among instances. The attention scores obtained from different heads, $n_h$ is a total number of heads, are concatenated, and a linear transformation is applied to ensure that the resulting shape matches the input, resulting in $\mathbb{R}^{n \times d}$ (eq: 5). To further process this contextualized information, we employ LSTM, which uses gating mechanisms and outputs the hidden layer of the last time step $\mathbb{R}^{1 \times d}$.

$$a_i^{self} = \text{softmax}\left(\frac{Q_i K_i^T}{\sqrt{d_k}}\right) V_i \tag{4}$$

$$A^s = (a_1^{self} \oplus a_2^{self} \oplus \ldots \oplus a_{n_h}^{self}) W_o \tag{5}$$

where $Q_i = HW_i^Q, K = HW_i^K$, and $V = HW_i^V$, for the $i^{th}$ head, are derived using trainable parameters $W_i^Q, W_i^K, W_i^V$, and $W_o$ linearly transforms the multi-head outputs. $d_k$ is used for scaling to prevent the dot product from becoming too large, and $C^s$ is the bi-LSTM processed output context vector from Branch 2 on $A^s$.

Furthermore, we use softmax normalized learnable parameters $s_0$ and $s_1$ to adaptively aggregate contributions from each pipeline's output. A scaling learnable parameter $\gamma$ fine-tunes the overall merged output contribution, introducing an additional degree of freedom in the weighting process (eq: 6). Inspired by attention principles, this approach facilitates contextual understanding and dynamic weighting for effective information extraction from both branches. It draws parallels from a multiple-layer fusion of contextual embeddings in ELMO during downstream tasks (Peters et al., 2018).

$$logits = \gamma \left(s_0 C^g + s_1 C^s\right) \tag{6}$$

After applying the adaptive aggregation method, a binary classifier with a single neuron and a sigmoid activation function is used to estimate the probabilities, $y$, of a slide being

positive. Subsequently, binary cross-entropy loss is computed at the slide level (Branch 1 and 2), while Smooth SVM loss (Lu et al., 2021) is applied for instance-level clustering (Branch 3). The Smooth SVM loss, a generalization of traditional cross-entropy classification loss, accommodates diverse margin values and temperature scaling strategies, providing flexibility to mitigate overfitting. The rationale for choosing Smooth SVM loss lies in addressing potential noise in pseudo-labels, offering robustness in the presence of uncertainties. The total loss, as per Equation 7, is calculated as the weighted sum of both losses, where $H'$ and $A^{g'}$ are the subset of $H$ and $A^g$ respectively, $\hat{y}$ is the ground truth, and $\beta$ is a hyper-parameter.

$$J = \beta \, \text{BCE}(y, \hat{y}) + (1 - \beta) \, \text{Smooth-SVM}(H', A^{g'}) \tag{7}$$

## 3. Experiments and Results

### 3.1. Experiment Setup

For a robust evaluation of classification performance, we employed 10-fold cross-validation. All methods were implemented in PyTorch and trained on a single NVIDIA RTX 3080ti GPU. The patch size for the YOLOv4-based glom detector was set to $6000 \times 6000$, and the MIL training involved 50-200 epochs with early stopping. $n_h = 4$, $\beta = 0.8$, a Bi-LSTM hidden dimension of 512, and Adam optimizer with $lr = 1e4$. Batch size is set to 1 for all models. Our code is available on GitHub [2].

### 3.2. Results

Table 1: Comparing our proposed model (LupusNet) with baselines, averaging results (in %) over 10-fold cross-validation on test cohort. Input types include GP (Only Glomeruli Patches) and AP (All Patches).

| Model | Input | Test AUC | Test F1 | Test ACC |
|---|---|---|---|---|
| ResNet-101 | GP | $52.88 \pm 20.54$ | $44.12 \pm 23.01$ | $53.23 \pm 18.26$ |
| Vanilla ViT | GP | $67.00 \pm 19.22$ | $56.96 \pm 23.58$ | $62.22 \pm 18.14$ |
| Max Pool | GP | $81.00 \pm 16.12$ | $72.82 \pm 16.04$ | $73.33 \pm 15.89$ |
| Average Pool | GP | $85.50 \pm 11.89$ | $76.98 \pm 17.30$ | $77.78 \pm 16.93$ |
| CLAM-SB | AP | $57.65 \pm 18.00$ | $52.22 \pm 15.34$ | $52.43 \pm 11.26$ |
| CLAM-SB | GP | $86.00 \pm 14.78$ | $72.80 \pm 12.66$ | $75.55 \pm 10.48$ |
| DSMIL | GP | $79.50 \pm 16.35$ | $68.34 \pm 17.98$ | $71.11 \pm 16.08$ |
| TransMIL | GP | $54.50 \pm 25.73$ | $50.11 \pm 24.17$ | $54.44 \pm 22.08$ |
| **LupusNet (Ours)** | **GP** | **$91.00 \pm 08.91$** | **$77.30 \pm 06.80$** | **$81.11 \pm 05.36$** |

We established baselines using a pseudo-labeling approach for lack of detailed glomerulus-level labels by assigning whole slide labels to all glomeruli and tested models like AlexNet, ResNet, and DenseNet, with ResNet-101 performing best (Table 1). These experiments

---

2. Code: https://github.com/CancerDiag/LupusNet

underscored the challenge of label inconsistency among similar glomeruli in lupus classes 4 and 5, affecting model accuracy and emphasizing the need for alternative methods in the absence of precisely labeled datasets.

Furthermore, we employed an end-to-end vanilla Vision Transformer (ViT) (Dosovitskiy et al., 2020) on glomeruli patches followed by weakly supervised max-pooling, average pooling, CLAM single-branched variant (CLAM-SB), DSMIL (Li et al., 2021) and Trans-MIL (Shao et al., 2021) and our proposed LupusNet on the in-house dataset. CLAM-SB results are presented for both scenarios, wherein we either input all the WSI patches or just the glomeruli patches. Pooling methods showed competitive performance, with max pooling achieving 81.00% AUC, 72.82% F1 score, and 73.33% accuracy, and average pooling resulting in 85.50% AUC, 76.98% F1 score, and 77.78% accuracy. The conclusive findings, as shown in Table 1, demonstrate that LupusNet outperforms all baseline models. We can empirically observe a significant performance improvement when only glomeruli patches are provided, consequently reducing noise to the weakly supervised models. Additional observation showed LupusNet outperforming CLAM-SB (GP), by a significant F1-score improvement for class 5 LN (65.17% to 77.03%), highlighting its efficacy in distinguishing the two classes, reducing false positives and enhancing precision.

## 4. Ablation Study

Table 2: Ablation study with module variations.
**L**=LSTM; **G**=Gated Attention; **C**=Clustering (Instance level)

| Model | Test AUC | Test F1 | Test ACC |
|---|---|---|---|
| LSTM | 64.00 ± 15.77 | 56.27 ± 9.70 | 60.00 ± 10.73 |
| L+G | 81.65 ± 13.90 | 67.00 ± 16.32 | 71.11 ± 11.94 |
| L+G+C | 85.00 ± 12.24 | 74.91 ± 15.56 | 77.78 ± 12.83 |
| ViT + G + C | 73.00 ± 18.01 | 64.99 ± 13.76 | 66.67 ± 12.05 |
| **LupusNet (Ours)** | **91.00 ± 08.91** | **77.30 ± 06.80** | **81.11 ± 05.36** |

In our ablation study, we methodically introduced various architectural components to evaluate their individual and combined effects on the model's performance. Beginning with a basic LSTM model as our starting point, we then integrated Gated Attention and Instance-level clustering. Each addition led to noticeable improvements in performance, as shown in Table 2, with our final model, LupusNet, outperforming all other configurations. This step-by-step process helped us identify the specific contributions of each component to the model's overall effectiveness in classifying two LN classes. We further optimized LupusNet by adjusting the learning rates and the number of Multi-Head Attention (MHA) blocks (Figure 3).

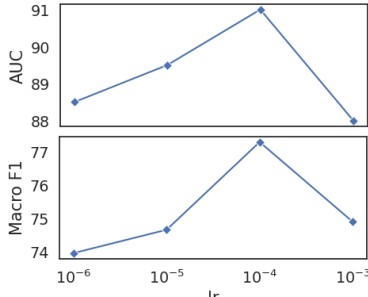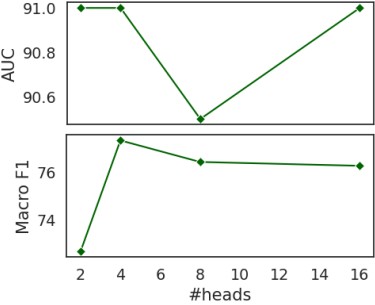

Figure 3: Hyperparameter tuning of LupusNet based on the optimized value of learning rate (left) and number of attention heads (right)

## 5. Discussion and Conclusion

Our study has showcased the application of MIL for LN subtype classification, which uses only slide-level labels, eliminating the necessity for glomeruli-level labels. Our idea was to explore how weakly-supervised methods perform in this situation and propose a framework (LupusNet) to improve it. Although using transformer-based models seems like a natural choice for their advanced context sensitivity, their empirical efficacy was suboptimal due to the reduced regions of interest. However, we recognized the need for self-attention among glomeruli for context inclusion. Therefore, our work includes this aspect without increasing network complexity by using LSTM and MHA. Furthermore, the attention weights can be assessed to infer the contribution of each glomeruli in the final classification which can also help reduce the inter and intra-variability among pathologists. Additionally, it holds significance for researchers studying other diverse renal diseases beyond the specific focus on LN. It also contributes to renal pathology research by creating a digital whole slide image dataset. While LupusNet exhibits promising results, there are areas for potential improvement. Our future work involves improving glomeruli detection models and feature aggregators, which could extract even better contextual information from glomeruli.

**Data Availability Statement:** The dataset generated and/or analyzed during the current study is available from the authors within the terms of the data use agreement and compliance with ethical and legal requirements (if any).

## Compliance with Ethical Standards

Procedures in studies with human participants adhered to ethical standards set by institutional (NIMS) and/or national research committees (ICMR).

## Acknowledgments

We acknowledge IHub-Data, IIIT Hyderabad (H1-002) for financial assistance. We also thank Dr. Manasa Kondamadugu for project coordination, Ms. Ramya Alugam, and Mr. Akula Rajesh Goud for data digitalization and organization.

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
