# OpenReview forum: "Lupus Nephritis Subtype Classification with only Slide Level Labels"
_MIDL.io/2024/Conference — MIDL 2024 Poster_

### Official Review · Reviewer_yXj5 · 2024-02-18

**Confidence:** 4
**Preliminary Rating:** 3
**Recommendation:** Poster
**Final Rating:** 5

**Summary:**

The authors present a dataset and method for lupus Nephritis subtype classification from WSIs, using a multiple instance learning paradigm. In particular, they show that predetection of glomeruli regions is really helpful for this task. The results indicate that the approach introduced by the authors outperforms standard MIL approaches such as CLAM.

**Strengths:**

- The authors specifically motivate their pipeline that includes a predection of glomeruli regions with the prior knowledge from histopathology, which makes a lot of sense, and, as the authors show, also works well. In fact, this contributes the major leap in AUC over the CLAM approach, as shown in Table 1.
- The authors provide an ablation study with multiple steps.
- The authors claim to make available the data, which helps reproducibility.

**Weaknesses:**

- The approach seems a bit complicated, especially in the combination of two long-range dependency modeling schemes (MHSA and LSTM). Could the authors explain why a more simple aggregation was not trialed? In particular, the ablation study suggests that the LSTM in itself features only marginal performance (AUC of 0.64, Test F1 of 0.56).
- The results given by the authors only report single shot performance. However, given the size of the dataset, we could expect a quite significant data dependency between patients. I think that the authors should run a cross-validation on their data and report mean+std (or results across runs) for a more fair comparison across approaches.
- The dataset is only available "upon reasonable request". If there is no distinct legal requirements against this, then I would strongly suggest to make the data available openly without preconditions, especially since it is listed as core contribution of the paper.
- The interpretability of the approach is questionable.

**Detailed Comments:**

- In histopathology, many datasets have been made available publicly already, given that anonymity is not a major concern for tissue images. Unfortunately, data that is only available "upon reasonable request" is often not available, or comes with restrictions that prohibit reproducibility. If there are legal obligations that prohibit the publications, please briefly comment on why this is the case, and to what degree these are not applicable when researchers request the data from the authors later. As a side note: The website that the authors have cited looks a bit unfinished and has a funny typo ("brian" instead of "brain"), making it look a bit unprofessional.
- I honestly did not get Figure 3. The weights seem to be very similar in Branch 2, and also I wonder what I would take home from these. Most importantly, I do not think that this in itself makes the approach interpretable. Investigating a single condition is also a weak spot in this. I suggest to remove the claim of interpretability.
- Please consider running an ablation study starting from CLAM (G+C) or just using gated attention-based MIL (G) as the base condition, rather than from the LSTM as base condition. And please always report mean +/- std deviation of consecutive runs or cross-validation, as the findings might else be just spurious.

**Justification Of Final Rating:**

I think the paper has seen significant advances in the rebuttal period and the authors were really responsive and reacted constructively to the criticisms. I also read the insightful comments of reviewer svAa and agree with the other reviewer and I am now also sure the paper meets the acceptance criteria for MIDL.

**Justification Of The Preliminary Rating:**

The approach is interesting and, especially the part about the preselection of ROIs, well-motivated. However, the paper is a bit on the complex side and not easy to comprehend, and the evaluation is not comprehensive.

**Questions To Address In The Rebuttal:**

- How robust are the advances over the CLAM architecture (with glomeruli preselected)?
- To which architectural contributions can these be attributed?
- Reasons why it is not possible to make the dataset available, or, better: a link to the publicly available dataset.

---

> ### Author Response · Authors · 2024-03-17
>
> We would like to start by thanking you for your thoughtful feedback. We are encouraged to learn that you've found our work to be of significant importance. We would now like to address all your queries one by one.
>
> 1. **Choice of combining 2 long-range dependency modeling schemes and why a simple aggregation was not trialed:** \
> The decision against a simpler aggregation method stems from our objective to address both the complexity and variability inherent in WSIs. While simpler aggregation methods can offer computational efficiency, they often fall short of capturing the detailed context and interdependencies between instances within a bag, which are crucial for the accurate classification of our task. \
> The ablation study with LSTM was conducted to demonstrate the incremental value each component (LSTM and MHSA) adds to the model. This experiment underlines the LSTM's role in enhancing the model's ability to process sequential patterns, as well as MHSA's ability to emphasize the important ROIs, resulting in a more effective model.
>
> 2. **Cross-Validation of the data:** \
> We have conducted all our experiments with 10-fold cross-validation, as mentioned in section 3.1. We would like to extend our sincere apologies for an oversight in our reporting which does not share the standard deviation. We acknowledge that this information is crucial for comprehensively understanding the variability and confidence in the performance metrics across different folds. The following are the results with respective mean and standard deviations for your reference. We will rectify this oversight in the next revision of our manuscript.
> Model | Input | Test AUC | Test F1 | Test ACC |
> -|-|-|-|-|
> ResNet-101 | GP | 52.88 ± 20.54 | 44.12 ± 23.01 | 53.23 ± 18.26
> CLAM-SB | AP | 57.65 ± 18.00 | 52.22 ± 15.34 | 52.43 ± 11.26
> CLAM-SB | GP  | 86.00 ± 14.78 | 72.80 ± 12.66 | 75.55 ± 10.48
> **LupusNet** | **GP** | **91.00 ± 08.91** | **77.30 ± 06.80** | **81.11 ± 05.36**
>
> 3. **Dataset Availability** \
> We appreciate your recommendation and understand the importance of open data access. We are currently working with our healthcare partners and the data foundation team to release the dataset in an appropriate manner, backed by the required infrastructure to host such a large image dataset. We will remove the phrase “upon reasonable request” in the next revision of our manuscript, as prompted by your reasoning. \
> To access the dataset, users need to register on the portal and agree to the usage policy. There will be no preconditions or formalities except for a very minimalistic inquiry form, which is standard practice for online data repositories. This form will be designed to gather basic information about the researchers and their intended use of the data. This process ensures that the data is, in its true essence, open to use by anyone for academic purposes. By the time the portal gets ready, anyone interested can write to us to request access to the dataset.
>
> 4. **Interpretation Section:** \
> The main advantage of our model lies in its enhanced ability to accurately identify the LN5 class, as mentioned in section 3.2 (by a significant F1-score improvement from 65.17% to 77.03%) which is characterized by uniform features across glomeruli. We observed that the gated attention mechanisms often fail to capture this uniformity as they only focus narrowly on glomeruli-level features. In contrast, our model employs a self-attention mechanism that assesses features across multiple glomeruli, hence improving the performance. Since the addition of branch 2 adds more information about different gloms, which branch 1 and 3 alone were not able to capture, hence this newly added information is responsible for improved performance and figure 3 is trying to convey the same. We can improve this explanation in our next revision, although we also understand the limited understanding and intuition the interpretation section conveys. We can remove this section if it promotes any confusion or offers less interpretability value to our overall claims and logic.
>
> 5. **Robustness of our model and the respective architectural contributions:** \
> We have conducted all our experiments with 10-fold cross-validation, as mentioned in section 3.1. As there are no similar, openly available datasets, a direct measure of robustness is not possible.  We have also shared the standard deviations for all the results, which should further give insights into the model’s performance. \
> We wanted to eliminate the 2-step process, which most of the previous works have adopted, a direct correlation was infeasible. The results, shared in section 4, reaffirm our understanding of the importance of multi-head attention and LSTM along with the already established gated attention and instance-level clustering. A combination of these 3 modules is the major contributor to the robustness and enhanced performance of the methodology as compared to the previous methods.

---

> ### Comment · Reviewer_yXj5 · 2024-03-19
> **Answer to the rebuttal**
>
> I am a bit surprised to see that the authors opted to not revise their paper according to the comments. Please note that MIDL explicitly allows for revisions during the rebuttal phase.
>
> **Regarding point 1:**
>
> While I thank the authors for elaborating on their decisions, I uphold my main criticism, which is that the architecture is complex and we can’t clearly attribute parts of it to the success of the paper, due to the missing ablation study that also experiments with simpler aggregation methods. While I see the point that long-range dependencies are of importance and can be beneficial, this is for now just an assumption and not proven. Simpler aggregation mechanisms like majority voting or average pooling are surprisingly effective. To this end, it is especially unfortunate that the authors chose to run their ablation study always starting from the LSTM, and not also without the LSTM. Hence, the insights remain limited, unfortunately.
>
> **Regarding point 3:**
>
> If the dataset is not made available with the publication on MIDL, it is not a contribution of the paper, and I would kindly ask the authors to not name this as such in the paper. It is, however, a pity, as the authors also point out comparison is hard without availability of the data.
>
> **Regarding point 4:**
>
> Better performance in one subclass does not make the model explainable or interpretable. I also still don’t understand what Figure 3 wants to convey, unfortunately. Furthermore, I think that an illustration based on a single, potentially cherry-picked, example is not really helpful to support the statements.

---

> > ### Author Response · Authors · 2024-03-25
> >
> > Thank you for your detailed feedback and for highlighting opportunities for improvement in our paper. We appreciate the time you've invested in reviewing our work, and your insights have been invaluable in refining our study.
> >
> > Initially, we were under the impression that revisions were to be submitted at the end of the discussion period, and were not aware that modifications could be made during the rebuttal phase as well. We apologize for this oversight. Following your suggestions, we have now revised our manuscript accordingly and believe these changes significantly enhance its quality and impact.
> >
> > 1. We have incorporated an additional ablation study that explores simpler aggregation methods, including mean and max pooling. These experiments have provided compelling evidence supporting our original hypothesis, confirming the superiority of our proposed method even when contrasted with these simpler approaches. This addition not only addresses your valid critique but also strengthens our argument by providing a more robust empirical foundation.
> >
> > 2. We've made a sample dataset accessible to all immediately after they create an account on the Data Foundation portal (https://datafoundation.iiit.ac.in/datasets), without the need for prior approvals. To access the full dataset, users simply need to request access post-sign-up, subject to agreement with the terms of use. We believe this strikes a balance between accessibility and responsible data management. Our dataset can be found under the "India Pathology Dataset (IPD)" section, with the name "Lupus Nephritis".
> >
> > 3. We have taken your feedback on the interpretability and explainability of our model, and we have decided to remove the interpretation section from our paper. This decision was made to avoid any misunderstanding and to ensure that our findings are presented as objectively as possible.
> >
> > We hope that these revisions and explanations address your concerns adequately. We are eager to hear your thoughts on the updated manuscript and remain open to further suggestions. Our goal is to contribute meaningfully to our field, and your expert feedback is crucial in achieving this aim.

---

### Official Review · Reviewer_svAa · 2024-02-29

**Confidence:** 4
**Preliminary Rating:** 4
**Recommendation:** Poster
**Final Rating:** 5

**Summary:**

The authors propose two main contributions: 1) a fairly large-scale dataset of Lupus Nephritis and 2) and end-to-end pipeline for LN subtype classification. The method utilizes attention mechanisms to enable slide-level labeling to be sufficient for training their network.

**Strengths:**

1. Dataset contribution is significant. This is a fairly large, diverse, and well constructed dataset that will be of great utility to the community.

2. The proposed method represents the first end-to-end pipeline for LN subtype classification. This is actually significant, not a trivial choice (like many end-to-end neural network approaches which often underperform their two-stage counterparts), because it allows for the gated and multi-headed attention mechanisms to implicitly solve the glomeruli type issue, with the final output solving the LN classification. Because of this uniquely powerful localization ability of attention, the network only needs slide-level labels, massively reducing the human effort involved.

**Weaknesses:**

1. It is unclear why the authors combined an LSTM with a transformer style network. Then entire point of a transformer is the attention mechanism removes the need for a memory unit. The HOW is clearly explained in section 2, but the WHY seems unclear. See questions below. But this makes the method feel a bit like, "we threw everything but the kitchen sink" at it, it would be great to see this solved via a pure transformer.

2. The authors have a few great comparisons, ResNet-101, CLAM, and an LSTM. These latter two are particularly good ablations. However, it would have been really nice to see some other related works here, rather than just baselines. I understand since this is a custom dataset, this would be replicating their methods, which hopefully have open source code, but the authors reference previous two-stage approaches and having an easy reference point of how those perform would be very valuable.

3. Figure 3 and section 3.3 claim to speak to the interpretability of the authors' approach. However, there is no interpretation provided. Just, here is a single test sample and some activations. Okay, cool. What does it mean? Are there consistent patters from one image to the next? There is no interpretation here. To make space for this (or just remove it, because as is is not worth keeping), I would recommend shortening the details in section 2.2. You don't need the equation for a bi-directional LSTM, people can go look that up because it's not novel to this work.

**Detailed Comments:**

No detailed comments, just see the questions to address in the rebuttal and weaknesses 2 and 3.

**Justification Of Final Rating:**

The new experiments are fantastic! Comparison with ViT and ViT in place of LSTM + MHSA are EXACTLY what this paper needed. I also appreciate the removal of the half-thought-out interpretation claims. I change my rating to a strong accept. I know there is still some hesitancy from the other reviewers, but as a consistent authors and reviewer of MIDL I think this work definitely meets the bar of publication for this venue. I appreciate the other reviewers concerns about the complicated nature of the method, which is somewhat unavoidable (see my strengths section about attention and the weakly supervised nature of the approach as well as the authors new experiments on a vanilla ViT), and concerns about the usefulness of a weakly labeled dataset (but honestly this is pretty standard for slide datasets in general). But I would strongly urge the AC to consider this work for publication. This is a fairly large scale dataset contribution and a fairly novel technique that outperforms ViT by a wide margin, that alone justifies publication easily.

**Justification Of The Preliminary Rating:**

The authors are most of the way to a great paper, there are just a few weaknesses that should be addressed to really significantly strengthen the paper around 1) justifying the LSTM better (and perhaps even removing it), 2) Adding baselines previous SotA methods (likely two-step methods) that came before, and 3) Actually fleshing out the interpretation section.

**Questions To Address In The Rebuttal:**

Why do the authors choose to combine an LSTM with their transformer based architecture. Is the memory unit attempting to solve a memory issue of some kind, where because you're patch sampling, you need some notion of other patches? Could this possibly be solved with some sort of sparse (long-context) style transformer instead of the complexity and inefficiency of an LSTM?

**Special Issue:**

No

---

> ### Author Response · Authors · 2024-03-17
>
> We would like to start by thanking you for your immensely thoughtful feedback. We are encouraged that you've found our work to be of significant importance and you've correctly highlighted the importance of this large dataset as well as the significance of our approach that is aimed at reducing the laborious human effort that goes on with the 2-stage approaches. We would now like to address all your queries one by one.
>
> 1. **Why did we combine LSTM with a transformer-style network?** \
> We agree with the assessment that at the heart of transformers is the attention mechanism. The decision to combine LSTM + MHSA, however, was driven by a pragmatic assessment of our dataset's limitations, which is the small number of ROIs (glomeruli images), which are insufficient for data-hungry models like transformers. We experimented with a vanilla ViT alone, as well as employing ViT in place of the LSTM + MHSA block (branch 2) while keeping branch 1 (gated attention) and branch 3 (Instance level clustering) as they are. But, as shown in the results below, neither of these ideas worked for us. This is why we chose LSTM, which performed very well in solving our problem. We recognize the limitations of LSTM, and in our future work, we plan to explore better alternatives and design choices to enhance the performance even further.
> | Model | AUC | F1 | Acc |
> |-|-|-|-|
> |Vanilla ViT (alone) | 67.00 ± 19.22 | 56.96 ± 23.58 | 62.22 ± 18.14 |
> |ViT in place of LSTM + MHSA (Branch 2) | 73.00 ± 18.01 | 64.99 ± 13.76 | 66.67 ± 12.05 |
> |**LupusNet** | **91.00 ± 08.91** | **77.30 ± 06.80** | **81.11 ± 05.36** |
>
> 2. **Choice of comparison models:** \
> We are glad that you've found our comparisons and ablations to be useful. We also acknowledge the value of reporting the results on previous two-stage approaches for direct comparisons, however, since our curated dataset does not have any glomeruli-level labels, it was not possible for us to make this direct comparison. One of the primary motivations behind our work is to eliminate the current challenge of acquiring glomeruli-level annotations, a laborious task, thereby facilitating the construction of a model capable of achieving reliable performance utilizing only slide-level labels and eventually reducing the workload of clinicians. For detailed implementation, please check out our code which we have now made available on [Github](https://github.com/AmitSharma1127/LupusNet).
>
> 3. **Interpretation section** \
> The main advantage of our model lies in its enhanced ability to accurately identify the LN5 class, as mentioned in section 3.2 (by a significant F1-score improvement from 65.17% to 77.03%) which is characterized by uniform features across glomeruli. We observed that the gated attention mechanisms often fail to capture this uniformity as they only focus narrowly on glomeruli-level features. In contrast, our model employs a self-attention mechanism that assesses features across multiple glomeruli, offering a more comprehensive analysis. Hence, other patches are required here. Since the addition of branch 2 adds more information about different gloms, which branches 1 and 3 alone were not able to capture, hence this newly added information is responsible for improved performance and Figure 3 tries to convey the same. We can improve this explanation in our next revision, although we also understand the limited understanding and intuition the interpretation section conveys. We can remove this section if it promotes any confusion or offers less interpretability value to our overall claims and logic.

---

### Official Review · Reviewer_wdSY · 2024-02-29

**Confidence:** 4
**Preliminary Rating:** 2
**Final Rating:** 4

**Summary:**

The paper proposed an attention-based multi-instance learning model, LupusNet, that allows classification of two glomerular types using only slide-wise labels. The results show that the proposed method outperformed others in instance-wise glomerular classification. The ablation study also provides a comprehensive comparison of the different contributions of the design.

**Strengths:**

1. The authors collected a dataset for LN subtype classification, which benefits the community.
2. The proposed attention-based MIL method achieves good instance-wise classification using only slide-wise labels.

**Weaknesses:**

1. The motivation and clinical value of the method are insufficient. During data collection, instance-wise annotations were collected, and fully supervised learning methods have already well resolved the classification tasks. There is no clear technical question or clinical problem in the data collection and supervised learning classification. Why do the authors frame the question as weakly supervised learning, creating a non-existent barrier?
2. More state-of-the-art MIL-based methods should be compared to demonstrate the functionality and capability of the proposed method, besides CLAM, which was proposed 3 years ago.
3. More details of the pipeline are lacking, creating a barrier for pipeline replication.

**Detailed Comments:**

1. What is the size of the patches? How are glomeruli of different sizes captured in the same patch size?
2. What is the distribution of the two classes in the dataset at the patch level and slide level?
3. What is the bag size? How does the bag size influence classification performance? How can it be ensured that the bag has sufficient glomeruli with the slide label?

**Justification Of Final Rating:**

I appreciate all the efforts from the authors during the rebuttal session, which improved the quality of the paper and clarified its scope by (1) clarifying the importance of slide-wise predictions for LN subtype classes and the contribution of the slide-wise labeled dataset, (2) providing additional experiments from other baseline methods, and (3) making the source code available for replication. I also appreciate the other reviewers for highlighting the contribution of the dataset as well as the innovation of the methods. I have increased the score for these efforts. I have increased the score for these efforts.

However, the discussion during the rebuttal revealed several limitations of the current stage of the work. The main concerns remaining are (1) the use of a small-scale private dataset, which poses the risk of cherry-picked results that may not fairly demonstrate the capability and generalizability of the models. If the authors aim to position this paper as a benchmark, additional datasets are encouraged to be included. (2) The explanation and interpretation of the results from the proposed methods may not fully showcase the models' capabilities. It is unclear whether the improved performance is due to the model design with correct knowledge learned from different gloms or simply due to increased complexity and parameters in the architectures. Therefore, some intermediate representations (glom-wise) of the models might help the audience to believe the functionality of the models. It should also be noted that achieving semi-quantified activity/chronicity scores for WSIs requires accuracy at the glomerular level, which is the main reason I questioned the motivation and contribution of the paper.

Overall, the quality of the paper has improved after the rebuttal, prompting me to increase the score. All the discussions and additional experiments during the rebuttal should be included in the final version of the paper if it is accepted.

**Justification Of The Preliminary Rating:**

The motivation and contributions of this paper are insufficient, and the results need further comparisons with state-of-the-art (SOTA) methods to demonstrate the functionality and capability of the proposed model. Additionally, the paper lacks detailed information necessary for replication. Therefore, I recommend rejecting the paper for MIDL.

**Questions To Address In The Rebuttal:**

Please further illustrate the motivation and contribution of the proposed method, as well as address the questions in the weaknesses and detailed comments sections.

---

> ### Author Response · Authors · 2024-03-17
>
> We sincerely thank you for your feedback and review. We would now like to address each of your queries one by one.
>
> 1. **Motivation and Clinical Value**: \
> We have mentioned our work's motivation and clinical relevance in Section 1's last three points, highlighting its importance for the Lupus Nephritis (LN) classification. Our methodology leverages only slide-level labels, avoiding the intensive task of glomeruli-level annotation. This strategy aids in efficiently classifying LN subtypes, potentially enabling quicker and more precise clinical diagnoses. Unlike previous studies that mainly depended on a two-step, glomeruli-focused annotation process, our work proves that effective LN subtype classification is possible using only slide-level labels. We believe this underlines the significance and clinical applicability of our research. We will add more clarity on this in the manuscript.
>
> 2. **Clarification on Data Curation:** \
> It seems there was a misunderstanding regarding our data collection process. We did not curate instance-wise (glomeruli-level) annotations; we acquired slide-level labels only. This distinction is critical as obtaining instance-level annotations for whole slide images is prohibitively labor-intensive. Our approach addresses this challenge by leveraging weakly supervised learning to make efficient use of slide-level labels, a method already established and successful in the field. We will clarify this in our manuscript to prevent any further confusion.
>
> 3. **Other MIL-based methods comparison:** \
> We acknowledge the importance of benchmarking against the latest advancements. However, it is important to note that there are no established baselines for this specific task. Our approach, incorporating gated and self-attention mechanisms, is designed to use the medical insights from Lupus Nephritis (LN) classes 4 and 5, expected to boost performance significantly. While we initially thought models like Vanilla ViT, DSMIL, and TransMIL might not compare well due to their attention mechanisms and transformers' limitations with limited data, we agree that including these comparisons will solidify our findings, prompted by your feedback. \
> Below is a brief overview of the performance metrics comparing our method against other notable models, including a non-MIL-based model, Vanilla ViT (trained from scratch). We will add these results.
> | Model | AUC | F1 | Acc |
> |-|-|-|-|
> | Vanilla ViT | 67.00 ± 19.22  |56.96 ± 23.58  |62.22 ± 18.14 |
> | DSMIL | 79.50 ± 16.35  |68.34 ± 17.98  |71.11 ± 16.08 |
> | TransMIL | 54.50 ± 25.73  | 50.11 ± 24.17  |54.44 ± 22.08 |
> | **LupusNet** | **91.00 ± 08.91**  | **77.30 ± 06.80**  | **81.11 ± 05.36**|
>
>
> 4. **More details of the pipeline are lacking, creating a barrier for pipeline replication**.  \
> We value the feedback and acknowledge the need for transparency in replicating our work. Section 3.1 of our manuscript provides all essential details for reproduction and for detailed implementation, we’ll include the [Github repository link](https://github.com/AmitSharma1127/LupusNet) in the camera-ready version.
>
> 5. **What is the size of the patches? How are glomeruli of different sizes captured in the same patch size?**  \
> The glom detector uses 6000x6000 patches, while the MIL model's feature extractor processes 224x224x3 images, as outlined in sections 3.1 and 2.2, respectively. Images of glomeruli are resized to 224x224x3 for the extractor, which then produces a 1xd vector, with d being the feature dimension size, which is 2048 in our study.
>
> 6. **What is the distribution of the two classes in the dataset at the patch level and slide level?** \
> Section 2.1 details the dataset distribution, highlighting the class breakdown with 62 slides for LN4 and 53 for LN5. Our approach is independent of glomeruli count or type, as slide-level labels (LN4 or LN5) don't directly correspond to glomeruli-level distinctions (normal, sclerosed, hypercellular, etc.). A single WSI can contain various glomerulus types, hence the distribution of two classes at a patch level does not hold much value, which is why it's omitted.
>
> 7. **What is the bag size? How does the bag size influence classification performance? How can it be ensured that the bag has sufficient glomeruli with the slide label?** \
> Our method utilizes the features of multiple glomeruli to derive insights at the slide level, akin to how clinicians assess slides. In line with the Multiple Instance Learning (MIL) framework, the 'bag size'—the number of instances or glomeruli— isn’t a hyperparameter to play with. It should be noted that we haven't defined a strict minimum number of glomeruli that a WSI must contain for it to be processed. We extended the formulation of binary MIL to subtype MIL in which at least one glomerulus is mandatory for our analysis to proceed which contributes to a subtype class.

---

> > ### Comment · Reviewer_wdSY · 2024-03-19
> > **Reply to the rebuttal**
> >
> > I appreciate all of the additional experiments and illustrations provided by the author during the rebuttal. Here are my comments on the rebuttal:
> >
> > 1. Motivation and Clinical Value:
> >
> >        Originally, I thought the proposed method could achieve instance-wise glom classification by only using slide-wise labels during training and evaluating the instance-wise glom classification performance in the testing stage. While several works have already addressed instance-wise glomeruli segmentation and classification, only providing the slide-wise label and training a model to give slide-wise predictions are insufficient and lack value for clinical research. It does not provide “precise clinical diagnosis” when the method only achieves slide-wise coarse prediction, reducing the precision of the prediction. The general application of MIL in WSI is due to the difficulty in obtaining pixel-level or patch-level annotations for disease diagnosis, while the target classification precision is also at the patient level or slide level (like CLAM). Even then, CLAM can also use attention scores to highlight disease regions on WSIs to evaluate the functionality and capability of their MIL model.
> >        Therefore, not providing instance-wise classification on each glomerulus and evaluating the classification performance at the instance level further reduces the innovation and contribution of the proposed method.
> >
> > 2. Clarification on Data Curation:
> >
> >        As mentioned in #1, if the annotation is only slide-wise grading without instance-level labels, the dataset's contribution and clinical value are lacking, since previous studies have achieved glomerulus-wise annotation. On the other hand, all previous glomerulus-wise annotation datasets can directly generate slide-wise grading labels proposed in this paper.
> >
> > 3. Other MIL-based Methods Comparison:
> >
> >        Since the dataset is very small and the prediction is at the slide level, the results may fluctuate between different epochs. Providing an instance-level prediction (by inputting only one glomerulus patch into the bag each time) can offer a more comprehensive evaluation of the models. Additionally, providing the results from a two-step glom-focus baseline method is necessary to understand the gap between supervised learning and weakly-supervised learning.
> >
> > 4. Pipeline Details:
> >
> >        At the time of writing these comments, no code has been uploaded to GitHub, which remains a concern about the details of the code and replication.
> >
> > 5. Patch Size and Glomeruli Capture:
> >
> >        I appreciate the clarification.
> >
> > 6. Class Distribution at Different Levels and Bag Size:
> >
> >        I was thinking about the instance-wise (glomerular-wise) distribution of different classes and instance-wise labels in each bag.

---

> > > ### Author Response · Authors · 2024-03-22
> > >
> > > 1. A fully supervised setting is an ideal situation. If this is expected, then there is no scope or need for MIL work. However, it can be seen that in the last few years, MIL has become increasingly relevant in the histopathology domain due to limitation in always obtaining pixel or patch-level annotation [ https://doi.org/10.1016/j.compmedimag.2024.102337 ]. In our work, we have showcased the application of MIL in renal histopathology, which earlier relied on obtaining annotation at the glomeruli level. Our idea is to see how MIL methods like CLAM work in this situation and propose a framework to improve it.  MIL framework is relevant since not all glomeruli contribute to the slide-level classification, and glomeruli classes are different than the LN subtype classes - which is usually not the case with other histopathology images where at least an instance (patch) belongs to the slide-level class directly [ https://doi.org/10.1016/j.eswa.2018.09.049 ].  \
> > > Our solution is end-to-end without the need to do the intermediate task of predicting glomeruli labels (multiclass classification) and aggregating it for slide-level prediction. It can be argued that the obtained performance may not be sufficient for clinical diagnosis. An AUC of 91 and AAC of 81.11 in the LN subtype classification provide confidence for further improvements with more slide-level data. Further, this is the first work in the literature to do this classification, setting the benchmark for further improvement. \
> > > Our work does not attempt to establish a one-to-one relationship between glomeruli classes and slide-level labels (LN subtype classes). However, the attention weights of each glomeruli can still be assessed to infer the contribution of each glomeruli in the final classification.
> > >
> > > 2. We beg to differ with this view that the dataset is of no use without instance labels. The generation of relevant medical datasets itself is of value given that they are not available in the first place. We have not seen any data on lupus available today of this size with various subtypes. Our work shows that it is of use by AUC of 91 and AAC of 81.11 in the LN subtype classification. We are only showing an alternate way to predict and are not claiming that it is superior to supervised learning.
> > >
> > > 3. In this work, we have chosen weakly-supervised learning since the glomeruli label is not available. It is not clear what the reviewer is suggesting to do under this setting to predict multi-class labels for evaluations. The attention map shows the relevant glomeruli, which are confirmed by pathologists to be well-known morphological features for that particular class. A supervised framework will be an idle solution, but we are dwelling on questioning how well MIL works in our data setting, benchmarking it, and improving it.
> > >
> > > 4. Please check out our code, which we have now made available on [GitHub](https://github.com/AmitSharma1127/LupusNet).

---

### Meta-Review · Area_Chair_VS8a · 2024-04-04

**Recommendation:** Accept (Poster)
**Confidence:** 4

**Metareview:**

All reviewers found the proposed method to be novel and the results promising, therefore I recommend acceptance.

---

### Decision · Program_Chairs · 2024-04-05

Accept (Poster)